# The Effect of Owner-Managers' Personality Traits on Organisational Ambidexterity in the Context of Small and Medium-Sized Enterprises

José Andrade [1], Luis Mendes [2,*] and Mário Franco [2]

1   Advanced Studies in Management & Economics Research Center (CEFAGE-UBI), ISMT—Instituto Superior Miguel Torga, 3000-132 Coimbra, Portugal; josericardoandrade@sapo.pt
2   Advanced Studies in Management & Economics Research Center (CEFAGE-UBI), University Beira Interior, 6201-001 Covilhã, Portugal; mfranco@ubi.pt
*   Correspondence: lmendes@ubi.pt; Tel.: +351-275319619

**Abstract:** This empirical study aimed to analyse the influence of the personality traits of owner-managers in small and medium-sized enterprises (SMEs) on organisational ambidexterity (OA). Based on the existing literature, five hypotheses were formulated about the relationships between the Big Five personality traits and organisational ambidexterity. A second-order structural equation model was used with a sample of 224 Portuguese SMEs in the sector of information technology (IT), telecommunications, and audio-visual and IT consultancy. The results obtained suggest that the personality traits of extraversion, neuroticism (versus emotional stability) and conscientiousness have a significant influence on organisational ambidexterity. These results are consistent with the research and demonstrate that owner-managers' personality traits influence organisational ambidexterity in SMEs. Theoretical and practical implications are explored.

**Keywords:** personality traits; Big Five model; owner-manager; organisational ambidexterity; SME; effect; context





## 1. Introduction

The idea that firms able to explore and exploit simultaneously are more competitive is the basis of research on organisational ambidexterity [1]. In this context, the literature refers to two fundamental aspects: the first refers to the difficulty of conciliating those activities due to the contradictions contained therein [2], considering the context (connected to the firm's relationship with its environment, organisational characteristics, availability of and access to resources), which can unequally influence the activities of exploration and exploitation [3]. The second concerns how small and medium-sized enterprises (SME) articulate resources to exploration or to exploitation [4]. Different theoretical contributions have allowed for a consideration of different possibilities concerning the factors contributing to the development of ambidexterity in firms in general and SMEs in particular, focusing their attention on the organisational level, studying how they divide their attention between the structurally separate activities of exploration and exploitation [3], or considering how those activities are conducted contextually [5].

At the individual level, the literature has neglected the influence of decision-makers in SMEs with regard to organisational ambidexterity, despite all the theoretical support we find in related theories such as the Upper Echelons Theory [6], Behaviour Theory of the Firm [7], or the Theory of Dynamic Capabilities [8].

Indeed, owner-managers leading and directing according to market and customers' demands have a significant impact on the success and the competitiveness of their SMEs. They have been characterised as responsible for strategic decisions made according to market demands, clients, suppliers, innovation processes, and others [9,10]. Characterised

by a smaller structure and fewer resources, SME are greatly influenced by global transformations affecting the markets [11]. In the case of Portugal, their influence on the economy is decisive due to the number of jobs they can create, their capacity to introduce innovative products, or to provide greater social integration and stability through the workforce [12]. These attributes justified the choice of this type of company for this study, especially when considering SME' specificities [11]. In these contexts, an owner-manager's actions have repercussions in terms of alignment and adaptability processes according to their operating markets, aiming for competitiveness [5]. According to Hambrick and Mason [6], owner-managers' personal characteristics are reflected in strategic decision-making processes. Other theoretical contributions have emerged in the literature to structure the debate around the owner-manager's influence on developing the capacity of organisational ambidexterity [13–16]. However, none of these approaches sought to advance a more intrinsic perspective of the owner-manager (such as personality traits) and if these influence organisational ambidexterity in an SME context or not.

In the field of studies about personality, the Big Five model of personality [17] has gained consensus in the literature, particularly in the area of personality studies in psychology, and, more specifically, in the field of management studies. Moreover, studies examining the relationship between personality traits and company performance present interesting perspectives as a research area [18]. This study aims to go further, through relating the Big Five model to organisational ambidexterity. While not seeking a psychological analysis of owner-managers, this research adopts the perspective of Hambrick and Mason [6] concerning their basic theoretical arguments about the intrinsic psychological nature associated with owner-managers, applicable to developing organisational ambidexterity in SMEs.

Considering the reasoning above, the aim of this research is to discover if there is a significant influence of owner-managers' personality traits on organisational ambidexterity in an SME context. Following the question asked by O'Reilly and Tushman [19], "how is ambidexterity achieved?", this research added a new perspective: is there an individual dimension of organisational ambidexterity in SMEs? The literature over the years has focused on studying the influence of structural and contextual dimensions above all. This study adopts the perspective of Bonesso et al. [14] in considering the relationship between individual behaviour and organisational ambidexterity as well as the line of thought suggested by Raisch et al. [13] when referring to the notion that ambidexterity also manifests at the individual level.

To answer the problems raised above and to address the objectives defined for this study, the paper is structured as follows: Section 2 provides a general view of the relevant literature, discusses the main studies, and develops the hypotheses formulated. Section 3 explains the research methodology. Section 4 describe the different results and provides a discussion of these results. Finally, in Section 5, conclusions are outlined, underlying the main contributions and suggestions for future research.

## 2. Literature Review and Hypotheses

### 2.1. Research on Personality and SME Performance

The owner-manager's relationship with SME performance is reflected in the literature, especially regarding its influence on different organisational configurations. At the heart of this perspective is the influence of the owner-manager's personality, which is reflected in how the firm aligns and organises itself internally [20] and how it is perceived externally [21,22].

Among the different contributions in literature about definitions of personality, we can highlight the notions of Schneider et al. [23], for whom personality is the set of individual attributes that give form, structure, and consistency to people's behaviour over time and when faced with different situations; we can also highlight that of Funder [24], who refers to the concept as a set of structures and tendencies that reflect or explain characteristic patterns of an individual's thought, emotions, and behaviours.

Despite the complexity of the concept, the literature has provided different studies showing the influence of the owner-manager personality on SME performance [25,26]. Some refer to styles of leadership and management [27], innovation capacity [28], orientation towards internationalisation [29], or entrepreneurial orientation [30].

The literature also refers to how the owner-manager's personality goes beyond the individual level to reach the collective or organisational level. This issue can be summarised essentially through two perspectives. The first perspective, according to Hofmann and Jones [31], is based on evidence about the relationship between owner-managers' personality traits and the results of firms' performance [32], through the analysis of behavioural regularities at the collective level. These authors explain that those regularities, at the basis of structures, processes, and dynamics established in firms, are identified in observable and relatively consistent behaviours over time. The second perspective consists of a conjugation between organisational context and leadership mechanisms. Here, the literature identifies different types. Organisational context mechanisms are operationalised in firms through processes that act on human resource management, such as an organisational culture [33] or leadership [34]. These mechanisms guide and reinforce the firm's strategic options, also functioning as a determinant of firm performance [35]. Also, the relationship between family businesses, the personality of their owners, and organisational ambidexterity is an interesting field of study that explores how the individual characteristics of family business owners can influence the company's ability to balance exploration and exploitation, two essential elements for long-term success. Research has suggested that certain personality traits can affect a family business owner's willingness to take risks, experiment with new ideas, and adopt innovative practices. These factors, in turn, can influence the company's ability to be ambidextrous [25,26].

### 2.2. Organisational Ambidexterity and the Big Five Model

Organisational ambidexterity is a concept linked to company performance and refers to their capacity to manage, simultaneously, processes of exploration and exploitation [36]. Organisational ambidexterity is the ability of firms to manage exploitation and exploration processes simultaneously. According to March [37], exploitation is related to refinement, improvement, selection, and execution, whereas exploration is related to innovation, flexibility, discovery, risk-taking, variation, and research [37]. Exploitation processes are mainly related to how efficient and disciplined firms are in managing these processes and how they are aligned in terms of competencies, systems, and context with the demands of the markets and the needs of customers. Exploitation involves individual and collective competences and knowledge, which, by being combined and internalised, allow for incremental refinements in terms of technology, product, or service; adapting existing technology; and better responding to current client needs. Exploration processes are mainly related to how firms engage with opportunities, are flexible and autonomous, and adapt to them. Exploration generally emerges as a response to latent environmental trends through creating innovative technology, revolutionary new products, and new markets [38]. To be competitive, firms must be able to manage the trade-offs between exploitation and exploration, which means that both require resources that must be managed in a balanced way. From this perspective, organisational ambidexterity is also a dynamic capacity [13] associated with the firm's capacity to adapt to highly competitive markets [5]. Without this balance, firms can fall into the trap of overvaluing exploitation or exploration, which can compromise the company's future.

The literature has pointed out the existence of antecedents of organisational ambidexterity—namely, organisational culture or human resource management practices [39]. Organisational ambidexterity is also contextualised in the literature at an individual level, and various studies support this perspective. Mom et al. [40] suggest the existence of a behavioural orientation associated with owner-managers, which combines exploitation and exploration in a given period of time, and Good and Michel [41]

refer to owner-managers' cognitive capacities regarding options between exploration and exploitation strategies.

Regarding studies about personality, the literature has suggested a relationship between the personality of top managers and company performance. Research in this area generally falls within the field of organisational psychology and management. Some studies have examined how specific personality traits of leaders can influence organisational culture [42], decision-making [43], innovation [44], and other factors that impact business performance [45]. Research on personality has gained consensus in recent years through the Big Five model of personality [17]. The theoretical perspective on the Big Five model of personality supports that the individual is a system characterised by internal dynamics that cause variation between adjustable and stable components in relation to an individual's real situation [46]. Different studies have related the Big Five model to entrepreneurial behaviour, risk-taking, locus of control, attitudes, self-efficacy, or innovation [47].

The Big Five model is based on the taxonomy of neuroticism, extraversion, agreeableness, openness to experience, and conscientiousness [48]. This taxonomy, being the most commonly used and validated conceptualization [35], systematises an integrated personality description of how individuals describe themselves and others. This model is not based on a specific theory of personality but is a model that summarises, according to McCrae and Costa [17], a theory of personality traits, including different streams of research. A personality trait is defined as a consistent pattern that regulates an individual's action, thought, or feeling as a response to a stimulus [49]. The model incorporates individual variables that are distinguishable and organised dynamically, acting in an interaction with an individual's context, considering their experience, and it assumes four assumptions of human nature, summarising the perspective of personality traits: knowledge, rationality, variability, and pro-activeness [17]. Knowledge is the assumption that personality is a valid subject of scientific study; rationality is the assumption that individuals are able to understand themselves and others; variability means that individuals are different from each other, considering the psychological dimension; and pro-activeness refers to individuals as the centre of their actions, having control of their lives, and where personality is an active element in defining their life paths. Each of these five factors is bi-polar and includes various specific traits: extraversion versus introversion, agreeableness versus antagonism, conscientiousness versus lack of orientation, neuroticism versus emotional stability, and openness to experience versus closure to experience.

Neuroticism (versus emotional stability) is a factor of personality characterised by an individual's tendency to feel negative emotions, nervousness, depression, impulsiveness, anxiety, or tension [50]. Individuals with high levels of neuroticism are pessimists, with low self-esteem, and their surrounding environment is perceived as threatening and difficult. According to the literature, other facets associated with neuroticism are self-awareness or even irritability [48]. However, individuals with low levels of neuroticism are calm, stable, and optimistic [51]. They better cope with pressure, challenges, and adversity, which can be beneficial in dynamic working environments. They are also more tolerant of ambiguity and are self-confident [52]. Low neuroticism is generally associated with emotionally stable people who are less prone to impulsive or emotional reactions to challenges, which can contribute to more rational decision-making—aspects that can support exploitation. Leaders with a low level of neuroticism can contribute to creating a more positive work environment and are efficient, which can also support exploitation. People with higher levels of neuroticism generally have a greater aversion to risk [53]. Organisational ambidexterity, which involves the simultaneous management of risky exploratory activities and safer exploitation activities, can be harmed if owner-managers are risk-averse. A positive work environment is often associated with creativity and innovation [54], promoting participation, decision-making, teamwork [55], and the involvement of everyone in the development and innovation processes [56], facilitating both exploration and exploitation [57]. Owner-managers with a high level of neuroticism tend to react more intensely to stress and pressure, as well as be more resistant to change and have difficulty making rational

and considered decisions, especially in high-pressure situations, which can contribute to inhibiting organisational ambidexterity. Nevertheless, some of the literature mentions that owner-managers who excessively cooperate will have difficulty in obtaining the resources and/or objectives necessary for their position [58].

From the point of view of organisational ambidexterity (being the effect of exploration and exploitation activities carried out simultaneously), neuroticism does not reflect the essence of either exploitation activities or exploration activities since, in organisational ambidexterity, both of these reflect differentiated forms of organisational learning [33] and knowledge [59]. The former follows an atypical path in developing new knowledge, aiming for a given objective; the latter is achieved through past knowledge, which is consolidated and secure [14,60,61]. From the reasoning above, the idea of variability associated with exploration, as well as extending and refining existing competences associated with exploitation, is not reflected in the neuroticism dimension, and this can be a factor of the distortion of organisational ambidexterity as a factor of influence, as described by the literature about the Big Five model. Excessive variability, a typical element of exploration activities, also does not reflect the focus on the result to be achieved, which can also be a factor of tension in the individual—an aspect that also affects the relationship between neuroticism and ambidexterity. Therefore, the following hypothesis is presented:

**H1:** *Organisational ambidexterity is negatively and significantly related to owner-managers' level of neuroticism.*

Openness to experience (versus closure to experience) is a factor characterised by the individual's tendency to become involved in different interests. The individual feels a need to engage in a variety of vocational activities characterised by novelty, change, and new experiences [50]. Individuals with high levels of openness to experience are curious, original, imaginative, and seek new sensations [51]. In firms, this factor has a positive relationship with leadership [62], as well as with strategy or performance [63]. The literature also mentions that this factor is related to owner-managers who actively seek constant change and new experiences, accepting the risk inherent to the process of researching, experimenting, and the variability of environments [64]. This process associated with owner-managers is reflected in their capacity to adapt to countless challenging environments, characterised above all by divergent thought and receptivity to a wide set of stimuli. Owner-managers with great openness to experience consider different possibilities of action as a function of their capacity to interpret, quickly and effectively, a diversity of information that does not fit with the existing mentality, thereby considering different strategies [65]. However, owner-managers with a low openness to experience tend to direct their actions towards a more directive style, seeking stability, efficiency, and a greater tendency towards gradual change [66]. The literature also suggests that openness to experience may not act as a facilitator of organisational ambidexterity, mainly due to contextual factors, such as the business strategy, company culture, difficulty in adopting innovation practices, or lack of human and technical or financial resources [39]. From the reasoning above, the idea of owner-managers' capacity for accepting different possibilities of action and strategy according to their interest and attraction to discovery, experimenting, and risk, is not sustainable, from a theoretical point of view, as a factor of influence in ambidexterity, as described by the literature on the Big Five model. From the point of view of organisational ambidexterity, and from the arguments above, the variability of business environments allows for the openness to experience factor to support exploration activities in SMEs, and not exploitation activities, conditioning organisational ambidexterity. Given this reasoning, the following hypothesis is presented:

**H2:** *Organisational ambidexterity is not related to owner-managers' level of openness to experience.*

Extraversion (versus introversion) is a classic personality factor where the individual's social character and gregarious nature is more valued. Individuals are satisfied with

themselves and with life, valuing social networks and relations with others [51]. Other facets referred to in the literature as associated with extraversion are cordiality, sociability, assertiveness, activity, and seeking excitement or positive emotions [48]. Individuals with high levels of extraversion tend to be confident, sociable, assertive, and emotionally positive, whereas individuals with low levels of extraversion (or introversion) are shy and of few words [51].

These characteristics have an effect in the context of business. Extraversion is a factor associated with an owner-manager's ability to motivate others, both internally and externally. The relationship between openness to extroversion and organisational ambidexterity can be explored by considering how associated characteristics can influence exploration and exploitation activities—namely, adaptability capacity and the ability to accept change; the ability to influence and persuade, which can be beneficial for supporting innovative initiatives (exploitation); or a greater willingness to take controlled risks, a fundamental characteristic for innovation and new opportunities [50]. This factor spreads throughout the company, as the literature refers to the formation of contact networks as an element of information dissemination, identifying business opportunities and strategic orientation in firms and developing appropriate solutions for those opportunities [65]. An owner-manager's capacity to adapt to the volatility and rapid changes in a business environment is also related to their influence at the firm level, not only in the development of new ideas and new internal processes linked to innovation and work behaviour [67] but also in the development of new strategies that allow for adaptation according to the owner-manager's ability to take the initiative and persuade and influence others. Although the relationship between extraversion and firm performance needs greater clarification, a link with innovation seems to exist [18]. Therefore, from the point of view of organisational ambidexterity, involving simultaneous exploitation and exploration activities, and considering the adaptive nature of the learning processes associated with each, the following hypothesis is proposed:

**H3:** *Organisational ambidexterity is positively and significantly related to owner-managers' level of extraversion.*

Agreeableness (versus antagonism) is a personality factor characterised by the individual's readiness to be affectionate, nice, and trustworthy. They can also be described as friendly, kind, altruistic, generous, fair, and anxious to help others [58]. Other facets associated with this dimension are modesty, sensitivity, cooperation, or acquiescence [48]. This factor is closely related to the capacity for team working and interpersonal relationships. The individual reveals a tendency to avoid conflict and reveals beliefs related to the importance of work, avoiding leadership and preferring to be led [18]. In a business context, the literature states that agreeableness is associated with certain behaviour, leading to a culture of non-risk and stability, characteristic elements of exploitation activities [65]. These aspects tend to inhibit the capacity to be innovative when considering the different market demands [68]. However, it is important to highlight that organisational ambidexterity is related to the functional relations necessary for specific processes within organisations, such as people management and teamwork practices [69]. Lubatkin et al. [70] and Gibson and Birkinshaw [5] conclude that environments promoting processes of socialization and recognition, culture, and interpersonal relations help to encourage ambidexterity, supporting the agreeableness factor in owner-managers. For Chang and Hughes [71], contextual conditions can increase the quality of internal communication to create and improve current products and services. Andriopoulos and Lewis [72] found that, in small firms, context favouring the emergence of ambidexterity could serve to support internal communication processes, facilitating the elimination of impractical processes. Nevertheless, the literature does not support the idea that this factor can initiate the development of ambidexterity, but it only contributes to this performance. Therefore, considering the arguments above, the following hypothesis is proposed:

**H4:** *Organisational ambidexterity is not related to owner-managers' level of agreeableness.*

Finally, conscientiousness (versus lack of orientation) is a factor characterised by leadership capacity, planning, respect, self-discipline, and respect for norms and efficiency. This dimension covers issues such as responsibility, controlling impulses, and orientation. Individuals with a high level of conscientiousness have a strong sense of direction, self-discipline, and orientation towards results. They are also characterised as being organised, hard-working, and determined. This is also a factor associated with behaviour directed towards objectives, order, a sense of obedience, and the need to comply with rules [48]. Conscientiousness is often associated with characteristics such as discipline, diligence, responsibility, and goal orientation, but it can also lead to the adoption of better management and governance practices. This can be crucial when balancing exploration activities, which involve experimentation, with exploitation activities, which often require routine and efficiency. The literature considers this factor fundamental for motivation in an organisational context [73], and it also frequently emerges in the literature associated with structured strategic decisions and with an owner-manager's formal and personal structural mechanisms [40]. In the context of ambidexterity, the essence of exploitation activities lies in experimental activities, based on the owner-manager's existing knowledge, as well as their entrepreneurial capacity, especially in the context of SMEs [66]. This also involves activities searching for new routines, structures, and systems, where owner-managers make decisions according to the challenges arising from market needs and in response to technological opportunities [19]. As organisational solutions of strategic orientation in SMEs, owner-managers base themselves on the context, the result of the market's competitive dynamics, with the conditions that allow for the establishing of the functional relations necessary for specific processes within the organisation [38] and internal environments facilitating the culture of ambidexterity [70]. From the arguments above, the following hypothesis is proposed:

**H5:** *Organisational ambidexterity is positively and significantly related to owner-managers' level of conscientiousness.*

## 3. Research Methodology

### 3.1. Sampling

A significant number of studies on organisational ambidexterity seek to build their research hypotheses in relation to a set of indicators of firm performance. Adhering to this premise, we frame the study's hypotheses around the effect of owner-managers personality factors to discover if those factors influence organisational ambidexterity in SMEs.

The SMEs participating in the survey were randomly selected through the InformaDB database and also by using the support of business associations in the chosen sectors of activity. This phase aimed to assemble firms operating in Portugal and guarantee the same number of firms for each of the different sectors chosen so that there were no sectors with a significant difference in the number of firms in relation to other sectors. The final sample is composed of 1202 firms meeting the following criteria: up to 250 employees and with a turnover of up to EUR 50 million. The justification for choosing this type of firms for this study lies in the fact that the market of the sampled SMEs is characterised by a high degree of uncertainty and demand. In such a context, to be competitive, SMEs depend greatly on their capacity to adjust and adapt to new developments and opportunities—important aspects within the scope of organisational ambidexterity [74]. In this context, this study argues that SME performance in organisational ambidexterity is influenced by their owner-manager's personality. We can therefore expect the owner-manager role to be related to activities of planning and improving existing internal processes, as well as exploration and exploitation activities affecting ambidexterity [13].

### 3.2. Participants

A questionnaire was constructed aiming to obtain some demographic data about the participants (gender, age, academic qualifications), as well as information about their relationship with the company, and about the company itself (years in the company, company size). The results show that the majority of respondents were male (79%), 28% were between 20 and 40 years old, 53% were between 41 and 56 years old, and 20% were over 56 years old. Regarding qualifications, 45% of the respondents had completed post-secondary but not higher education, 38% had a degree—master's or Ph.D., and 17% stated they had completed secondary education (in its various forms) or less than this.

Concerning the respondents' position in the firm, 83% of the participants said they belonged to firms with up to 50 workers, and 17% belonged to firms with between 51 and 100 workers. Regarding time spent in the firm, 20% answered they had been there for a period ranging up to 20 years, 43% had been there for a period between 21 and 30 years, and 38% had been there for over 30 years.

### 3.3. Instrument

The data were collected through a questionnaire based on a set of scales already used and tested in previous studies (in a wide range of different contexts) and self-administered by owner-managers.

Before the administration process, the research questionnaire was subject to a validation process in three stages, which included professionals working in IT, telecommunications, top management functions, consultants in the area of SME management and organisation (in the first two stages), and four SME owner-managers in the telecommunications and management consultancy sectors (in the third stage). Each stage contributed to adjustments in the questionnaire in order to improve interpretation and the order of the questions. Some items were modified to better align with the specific context under study. The different dimensions used in this study belong to instruments already employed in earlier studies by other researchers. Because the scales were developed in English, we used the conventional translation method, i.e., translation of English to an original Portuguese version. The process used allows for time and cost efficiency.

However, this translation method depends on the translator's experience and knowledge, and can sometimes result in low levels of validity and reliability of the study instruments [75]. To limit this possibility, a translator, a native English speaker, was asked to confirm the translation from English to Portuguese. Thereafter, the questionnaire was analysed by a number of consultants and higher education lecturers for a validation of the process of translation to Portuguese, based on the dimensions of the original questionnaire. The process allowed for the refining of some questions in the Portuguese version to avoid any ambiguity or misunderstandings. Other improvements were reached after the pre-test to ensure that the questions were clear, relevant, and interpreted as expected.

Considering the well-known difficulty in managing paper questionnaires and the respective response, we decided to administer the questionnaire online, which was constructed and developed on an appropriate internet platform and following the recommendations of Dillman [76]. This type of approach ensured that all the items were answered, preventing any from being left blank.

The questionnaires were gradually sent out by groups; before this process, an attempt was made to contact the firms in each group to provide explanations about the questionnaire and the underlying research project. After this step, a questionnaire was sent by e-mail to each company, addressed to the owner-manager. In some cases, extra effort was made to encourage completion of the questionnaire through a direct telephone call. Here, it was explained to the participants that they would have access to a summary of the main evidence from the study. Of the 1202 questionnaires sent out, 224 were received and duly completed, representing a response rate around 19%.

### 3.4. Measurements and Scales Development

The study variables were operationalised through items on a Likert-type scale to obtain more reliable and valid results [77]. The aim of this study was to employ instruments already used and validated in other studies and with a good level of internal consistency.

### 3.4.1. Ambidexterity

The ambidexterity scale used in this study was developed by Lubatkin et al. [70], based on scales developed previously by He and Wong [78] and Benner and Tushman [3]. The scale proposed by Lubatkin et al. [70] includes twelve items, six of them reflecting the exploitation dimension and the other six reflecting the exploration dimension. The six items formulated according to the exploration orientation consist of statements such as "looking for creative ways to satisfy customers". Similarly, the six items formulated according to exploitation guidance consist of statements such as "searching for commits to improve quality and lower cost". Both dimensions are assessed on a seven point Likert-type scale, ranging from "Strongly Disagree" (1) to "Strongly Agree" (7), in order to ensure a statistically significant variability of the answers obtained. The participants were asked to indicate their level of agreement with the statements concerning activities carried out according to their influence on them as firm owner-managers.

### 3.4.2. Personality

To assess the owner-managers' personalities, we used the BFI-K instrument by Kovaleva et al. [79], a version of Rammstedt and John's Big Five Personality Inventory instrument [80]. According to Kovaleva et al. [79], this scale presents good reliability and validity. The BFI-K was chosen for this study for two reasons. The first one concerns the scale's potential to be applied to larger samples, considering its qualities, and the second one is because the BFI-K is an economical instrument for use in studies based on online questionnaires [79]. Personality questionnaires are usually extremely long, and it is also a challenge for the researcher to choose a personality measuring instrument that, while bearing in mind ideal research questions, focuses on practical aspects, such as the ease of answering and the time required to complete the questionnaire. The BFI-K [79] corresponds to these requirements.

The BFI-K [79] is a short scale that includes twenty-one items, which facilitates the respondent's participation, and is developed around five dimensions: extraversion (e.g., "I see myself as someone who tends to be quiet"), agreeableness (e.g., "I see myself as someone who is generally trusting"), conscientiousness (e.g., "I see myself as someone who does things efficiently"), neuroticism (e.g., "I see myself as someone who worries a lot") and openness (e.g., "I see myself as someone who is curious about many different things").

All the dimensions of the BFI-K are evaluated on a seven-point Likert-type scale from "Strongly Disagree" (1) to "Strongly Agree". All the dimensions are operationalised through four items, except for openness to experience, which is assessed through five items. The participants were asked to indicate their level of agreement with each statement related to personality characteristics.

### 3.5. Statistical Procedures

A structural equational approach was adopted to assess the influence of owner-managers' personality traits on organisational ambidexterity. This approach allowed for a better representation of the variables studied and also for the association of measurement errors with endogenous and exogenous variables, allowing for multiple indicators of latent constructs [81]. Here, we decided to represent organisational ambidexterity as a second order construct inasmuch as both exploration and exploitation are constructs intrinsic to organisational ambidexterity.

The methodology of Cao et al. [82] was applied in order to understand the level of ambidexterity achieved by SMEs in the sample. According to Cao et al. [82], the simultaneity inherent to the ambidexterity concept does not mean that both exploration and exploitation

reach the same level of intensity. Firms can be ambidextrous without having the same level of intensity, and this occurs at different moments according to the specific contingencies of the context and business environment the firm is part of.

This is an important process, since it allows us to consider the research design of this study as being supported theoretically. The approach adopted in this study was based on the methodology proposed by Cao et al. [82], which sustains this principle. The approach that follows considers a dynamic configuration of organisational ambidexterity, through the perspective of balancing ambidexterity and the combined perspective of ambidexterity. The former considers the exploration mean and the exploitation mean; the latter considers the product between them. Table 1 presents the level of ambidexterity achieved by SMEs in this study.

**Table 1.** Level of balanced ambidexterity and combined ambidexterity achieved.

| | Balanced Ambidexterity | | Combined Ambidexterity | |
|---|---|---|---|---|
| | (A) Average of Exploitation | (B) Average of Exploration | A*B | $\sqrt{A*B}$ |
| Level of Ambidexterity | 4.63 | 5.90 | 27.32 | 5.22 |

SMEs in our sample reveal a high balance of ambidexterity, since the levels of exploration and exploitation are high (considering that the maximum is 7), and a level of ambidexterity (combined view of ambidexterity) of 27.32, when the maximum possible is 49. The other indicators present values supporting this conclusion. Thus, the level of organisational ambidexterity reflects the ratio of the scale used (Likert-type from 1 to 7) through the square root of each of the means of each completed questionnaire, with the value of 5.22 being considerably high.

### 3.6. Concerns about Common Method Bias

After the initial validation to check the levels of balanced and combined ambidexterity, according to the methodology proposed by Cao et al. [82] and Dolz et al. [83], the next step was to validate the measurements of the structural equation model studied. Since all the information gathered in this research came from a single questionnaire, the recommendations of Podsakoff et al. [84] were followed regarding the variance attributed to the data-collecting method (common method bias).

Common method bias (CMB) occurs when variations in answers are caused by methods used rather than interviewees' attitudes. In this sense, the collection method biases the variations to be analysed. To test this effect, the Harman single factor test was used, in which all the items (measuring latent variables) are loaded on a common factor. A total variance for a single factor under 50% suggests there is no CMB that would bias the data. Therefore, to detect the presence of CMB, a factor analysis was applied with all the variables used in the model. One factor, without rotation, was extracted, and the result obtained captured only 16% of the variance; therefore, CMB was not considered a threat in this study.

### 3.7. Analysing Statistical Assumptions

The data obtained were analysed based on the biases for suspect response patterns, outliers, and answer inconsistencies. Concerning the first one, no missing data were detected; regarding the second one, the existence of outliers was assessed through the squared Mahalanobis distance ($D^2$), and the answers were analysed searching for patterns or the repetition of the same type of answer to different questions.

The assumption of variable normality was assessed through the univariate and multivariate coefficients of asymmetry (*Sk*) and kurtosis (*Ku*). No variable presented *sk* or *ku* values indicating severe violations of normal distribution. Asymmetry values (*Sk*) ranged between 0.027 and 0.943, and kurtosis (*Ku*) remained between 0.020 and 1.651, suggesting no violation of these assumptions, since both remained below the values indicated in the literature: $|Sk| < 3$ and $|Ku| < 10$ [85]. The KMO criterion was also used with the

classification criteria defined in Hair et al. [85]; the KMO value obtained was equal to 0.807. Regarding multicollinearity, we used the VIF and Tolerance values; VIF values were below 4.261 and Tolerance values were over 0.235, indicating a low level of multicollinearity [85].

## 4. Results

### 4.1. Validation of the Measurement Model

The literature recommends that the process of validating the research model should be performed in two phases: firstly, a factor validation of the measurement model, and secondly, a validation of the structural model. To validate the measurement model, a confirmatory factor analysis (CFA) was performed using the AMOS software (v.24), in order to adjust the model [77]. The maximum likelihood estimation method was used, because, as reported in the literature, this is the most common approach in structural equation modelling for its robustness [85]. The factor weights ($\lambda \geq 0.5$) were determined, and items with reduced individual reliability ($R^2 \leq 0.50$) were withdrawn. Items Extra03, Agree04, Consc01, Explor01, and item Exploit06 presented a considerably lower factor weight. The adoption of a more conservative perspective aimed for correlation between factors which, theoretically, should be orthogonal [85]. After analysing the modification indices, the Explor05 item was found to saturate in more than one factor; therefore, a decision was also made to remove it and have the model redone.

A reliability analysis of the measurement scales was performed through the Cronbach alpha, normally used in studies with constructs based on various Likert-type scales. The results observed (see Table 2) indicate suitable levels of internal consistency for all the scale variables used, varying between 0.798 and 0.911 [86].

**Table 2.** Factor analysis with the alpha of Cronbach coefficient values.

| Constructs | Items | Individual Reliability | Standardised Regression Weight Reliability | T-Values | Alpha Cronbach |
|---|---|---|---|---|---|
| Extraversion | I see myself as someone who is outgoing, sociable. | 0.74 | 0.87 | 4.49 | 0.83 |
| | I see myself as someone who generates much enthusiasm. | 0.61 | 0.78 | 7.22 | |
| | I see myself as someone who is reserved. | 0.51 | 0.71 | 8.42 | |
| Agreeableness | I see myself as someone who is generally trusting. | 0.73 | 0.86 | 5.04 | 0.80 |
| | I see myself as someone who can be cold and aloof. | 0.60 | 0.77 | 7.68 | |
| | I see myself as someone who tends to find fault with others. | 0.57 | 0.76 | 7.89 | |
| Conscientiousness | I see myself as someone who makes plans and follows through with them. | 0.76 | 0.88 | 4.19 | 0.83 |
| | I see myself as someone who does a thorough job. | 0.59 | 0.76 | 7.65 | |
| | I see myself as someone who tends to be lazy. | 0.53 | 0.73 | 8.21 | |
| Neuroticism | I see myself as someone who is depressed, blue. | 0.74 | 0.86 | 5.90 | 0.85 |
| | I see myself as someone who is relaxed, who handles stress well. | 0.60 | 0.77 | 8.17 | |
| | I see myself as someone who worries a lot. | 0.53 | 0.71 | 8.81 | |
| | I see myself as someone who gets nervous easily. | 0.52 | 0.70 | 8.83 | |
| Openness | I see myself as someone who is curious about many different things. | 0.69 | 0.83 | 7.25 | 0.87 |
| | I see myself as someone who has an active imagination. | 0.66 | 0.81 | 7.70 | |
| | I see myself as someone who has values artistic, aesthetic experiences. | 0.61 | 0.80 | 7.90 | |
| | I see myself as someone who is ingenious, a deep thinker. | 0.55 | 0.75 | 8.60 | |

**Table 2.** *Cont.*

| Constructs | Items | Individual Reliability | Standardised Regression Weight Reliability | T-Values | Alpha Cronbach |
|---|---|---|---|---|---|
| Exploitation | Your action has sought to focus on fine-tuning what it offers to keep current customers satisfied. | 0.82 | 0.90 | 5.91 | 0.91 |
| | Your action has sought to continuously improve the reliability of your firm's products and services. | 0.73 | 0.75 | 7.92 | |
| | Your action has sought to increase the levels of automation in your firm's operations. | 0.70 | 0.83 | 7.53 | |
| | Your action has sought to focus on commitments to improve quality and lower costs. | 0.56 | 0.80 | 9.16 | |
| | Your action has sought to focus on constantly surveying customer satisfaction. | 0.55 | 0.75 | 9.56 | |
| Exploration | Your action has sought to focus on looking for creative ways to satisfy customers' needs. | 0.72 | 0.85 | 6.79 | 0.89 |
| | Your action has sought to actively target new customer groups. | 0.65 | 0.81 | 7.79 | |
| | Your action has sought to create products or services that are innovative to the firm. | 0.61 | 0.80 | 7.96 | |
| | Your action has sought to bases the success of your firm on its ability to explore new technologies. | 0.59 | 0.77 | 8.42 | |

After removing the items and correlating the errors based on the modification indices proposed by AMOS, a good adjustment quality was obtained, except for GFI (although it was very close to 0.9, indicating a good model) ($X^2/df$ = 1.469; CFI = 0.957; GFI = 0.895; RMSEA = 0.045; PCFI = 0.810; PGFI = 0.691) [85]. Table 2 presents the values of individual reliability and the alphas for the construct, and Table 3 presents the assessment of the measurement model with the values of AVE, CR, MSV, and ASV.

**Table 3.** Assessment of the measurement model.

| Constructs | AVE | CR | MSV | ASV |
|---|---|---|---|---|
| Extraversion | 0.62 | 0.83 | 0.07 | 0.04 |
| Agreeableness | 0.62 | 0.83 | 0.06 | 0.03 |
| Conscientiousness | 0.60 | 0.86 | 0.31 | 0.10 |
| Neuroticism | 0.61 | 0.82 | 0.31 | 0.10 |
| Openness | 0.63 | 0.87 | 0.10 | 0.06 |
| Exploitation | 0.67 | 0.91 | 0.06 | 0.03 |
| Exploration | 0.66 | 0.91 | 0.10 | 0.05 |

*4.2. Validation of the Structural Model*

To study the structural model (Figure 1), the maximum likelihood estimation method was used to determine the model's adjustment indices. With the model adjusted through the modification indices (above 11; $p < 0.001$), produced by AMOS, and based on theoretical elements, the following indicators were obtained, confirming the model's adjustment and consequent internal and external consistency: $X^2/df$ = 1.571; CFI = 0.946; GFI = 0.889; RMSEA = 0.051; PCFI = 0.815; PGFI = 0.694) [85].

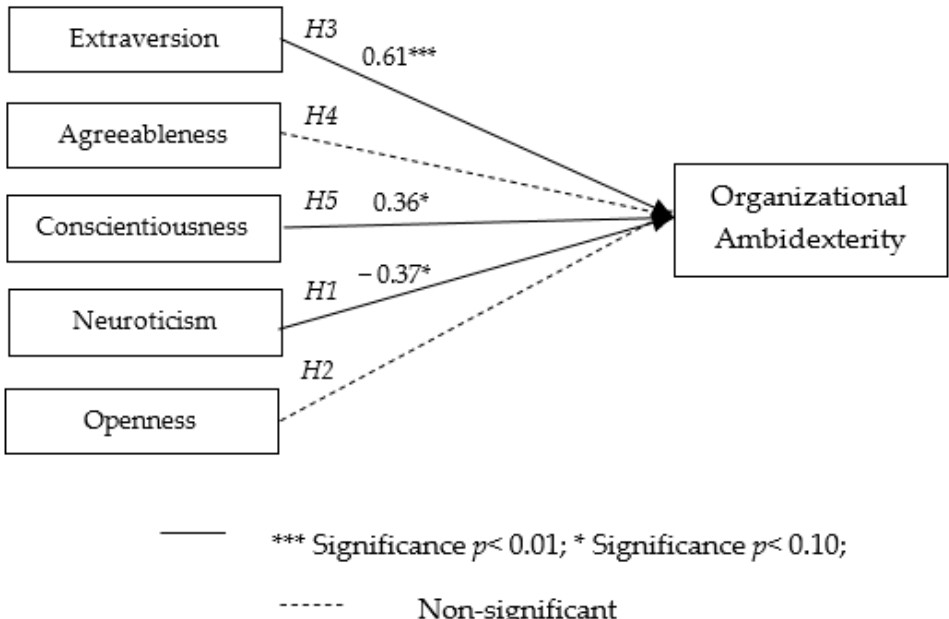

**Figure 1.** Structural model.

This study analysed the relationships between personality traits according to the Big Five model—neuroticism, openness to experience, extraversion, agreeableness and conscientiousness, and organisational ambidexterity. The measurement model of the latent factors explains 61% of the variability of the Big Five model regarding organisational ambidexterity. The paths analysis between the factors revealed that Extraversion- > OA presents the greatest weight ($B_{\text{Extra·OA}}$ = 0.193; *SE* = 0.060; $\beta_{\text{Extra·OA}}$ = 0.610; *p* = 0.001), followed by Neuroticism- > OA ($B_{\text{NEURO·OA}}$ = 0.122; *SE* = 0.072; $\beta_{\text{neuro·OA}}$ = −0.366; *p* = 0.090) and Conscientiousness- > OA ($B_{\text{consc·OA}}$ = 0.105; *SE* = 0.056; $\beta_{\text{consc·OA}}$ = −0.360; *p* = 0.059). The paths Openness to Experience- > OA and Agreeableness- > OA are not significant ($B_{\text{OPEN·OA}}$ = 0.034; *SE* = 0.057; $\beta_{\text{OPEN·OA}}$ = −0.116; *p* = 0.547; and $B_{\text{AGREE·OA}}$ = 0.011; *SE* = 0.060; $\beta_{\text{AGREE·AO}}$ = −0.039; *p* = 0.856). According to the model summarised in Figure 1, data obtained gave support to the five guiding hypotheses for this research.

## 5. Discussion and Conclusions

### 5.1. Summary of the Results

This study aimed to show the influence of personality traits on organisational ambidexterity in Portuguese SMEs from the sector of information and computing technology, programming, telecommunications, and audio-visual and IT consultancy. Before this study, as far as we know, no studies had focused on this type of influence on OA using a second-order structural equation, modelling for an ambidexterity variable, and it corroborates some of the literature on the influence of owner-managers' personal characteristics on the performance of SMEs and organisational ambidexterity. The results obtained suggest that organisational ambidexterity is positively and significantly related to the personality traits of extraversion, neuroticism (versus emotional stability), and conscientiousness but not with agreeableness and openness to experience. These results are consistent with the literature [18,65,87], inasmuch as these studies also report the influence of personality traits on firms' performance variables, but not all of these traits influence these variables in the same way. The explanation may lie in the fact that these factors may not related to just exploration or exploitation but rather with both.

*5.2. Contributions for Theory and Management*

The research path defined for this study helps to understand and updates knowledge about the influence of owner-managers' personality traits on organisational ambidexterity through the Big Five model. This perspective, based on the development of OA by the owner-managers of Portuguese SMEs, is built on their behavioural orientations towards exploration and exploitation.

By adopting the perspective of Bonesso et al. [14] regarding the perceptions of owner-managers for an ambidextrous performance, this study contributes to advancing this area of research. With this in mind, this study has brought two new perspectives to the literature. The first sheds light on the specific reality of Portuguese SMEs by suggesting that extraversion personality traits, conscientiousness, and a low level of neuroticism characterise the Portuguese management style relating to Portuguese SMEs. The second perspective highlights the importance of the Big Five personality model as an explanatory model for organisational ambidexterity. With these aspects in mind, this study brought several new features.

At the theoretical level, the study clarified and updated knowledge related to owner-managers' characteristics in forming organisational ambidexterity, based on other studies with similar objectives. Thus, for example, the contribution of Mom et al. [40], concerning the individual perspective of ambidexterity, maintains its original limitation, as it does not support the owner-manager's individual characteristics as being at the basis of behavioural orientations towards ambidexterity. This study is also in line with what some research has already recommended, especially when relating cognitive styles to exploitation and exploration. Thus, De Visser et al. [88] state that those with a more analytical cognitive style tend to be related to exploitation, while more intuitive cognitive styles are closer to the exploration of new products and markets, which can help clarify some of the results obtained with regard to exploration and exploitation.

Hence, for the first personality trait, extraversion, the literature does not support a specific relationship with exploration or exploitation. However, extrovert owner-managers are essentially able to influence others positively. Extraversion is supported in the literature in different studies on ambition, orientation towards objectives, work, and leadership or effectiveness, elements that are supported in the literature on ambidexterity in relation to exploration and exploitation activities, which, from the theoretical point of view, is sustainable in our study.

Concerning the neuroticism (or the level of emotional stability) factor, here too the literature fails to present a relationship between this factor and exploration or exploitation activities. The literature states that a significant part of owner-managers are emotionally stable, and this stability is reflected in how they manage their firms [18]. From this perspective, owner-managers with low levels of neuroticism are optimistic, entrepreneurial, have a positive level of self-efficacy, feel less threatened by uncertainties in the business environment, and have an adaptive view according to the need to change. These elements also appear in the literature on ambidexterity, which corroborate the conclusions of this study when related to exploration and exploitation activities. Nevertheless, the levels of neuroticism found in this study were negatively associated with ambidexterity. The mean obtained from all the completed questionnaires is 3.59, which, to some extent, explains the results reached, inasmuch as there is no clear definition of low neuroticism or high emotional stability. Here, the results obtained corroborate those of other studies [18], suggesting that the neuroticism factor is indeed negatively associated with organisational ambidexterity.

For the conscientiousness trait of personality, the results obtained in this study suggest a positive relationship with organisational ambidexterity, corroborating previous studies. In the literature, both exploration and exploitation are activities requiring a focus on the results to be achieved, on seeking positive performance, efficiency, and variability. In this context, the conscientiousness factor emerges in this study in line with others by also reporting a positive relationship with performance results in different types of work [49],

as well as a positive relationship with innovation. Owner-managers with high levels of conscientiousness are generally characterised as individuals with a strong sense of responsibility, discipline, and the will to follow rules and procedures. Despite the results achieved, we must also consider that there is no evidence no show whether the relationship established between exploration and exploitation is orthogonal or not. Further studies are therefore needed to explore this concern.

The hypotheses proposed in this study, concerning the absence of the influence of the openness to experience personality trait, were also confirmed, since the results obtained were non-significant, which follows in the direction indicated by some of the literature, not without some ambiguity. McCrae and Costa [51] suggest that openness to experience is a factor reflecting individuals' tendency to seek change scenarios, and Nadkarni and Herrmann [65] emphasise that idea that this factor can be linked to firm performance. However, Gow et al. [18] argue that owner-managers with high levels of openness to experience are related to more innovative firms. Indeed, despite some studies in the literature suggesting a positive relationship between openness to experience and, for example, innovation, this study did not show a positive effect in relation to organisational ambidexterity in the SME context. In this sense, it is important to mention that openness to experience is a personality trait closely related to the need for change, experimentation, and discovery, and in the literature on ambidexterity, these elements are associated with exploitation—but not exploration—activities, which may help explain the results obtained in this study with this personality trait. Other aspects may also explain why openness to experience is not related to organisational ambidexterity. At this level, the literature refers to an organisational culture that does not value innovation; the scarcity of technical, human or financial resources; and decision-making [11]. Another aspect that can condition the relationship between openness to experience and organisational ambidexterity is the strategy adopted by the organization, including its orientation towards innovation and efficiency [89], which can affect the relationship between openness to experience and ambidexterity. In addition to the information above, it should be noted that organisational ambidexterity is a combination of exploration and exploitation, and other personality traits such as conscientiousness and neuroticism can also influence the relationship between openness to experience and organisational ambidexterity [90].

Finally, concerning the agreeableness factor, the results obtained are equally non-significant. These results are supported by the research, since the literature states that this factor is not associated with leadership [65] or with factors such as performance [91], also suggesting a reduced tendency for this factor to be associated with innovation processes, investment risks, or aggressive business strategies [18]. In this sense, it is important to highlight that the literature states that both exploitation and exploration mean different behaviours associated with different cognitive processes. While, in exploration, cognitive processes are associated with the search for information, learning, experimentation, and openness to novelty, in exploitation, cognitive processes are associated with refining knowledge, the efficient execution of known tasks, and the consistent application of acquired knowledge [92].

This study also contributed to the Upper Echelons Theory, advancing it a little more by suggesting that the relationship between owner-manager characteristics and SME performance towards organisational ambidexterity can be influenced by personality traits, at the individual level, and in line with other studies [65]. In addition to this issue, our study also shows the importance of the Big Five model in contributing to organisational ambidexterity in an SME context. As far as we know, no other study has sought to relate the Big Five model with organisational ambidexterity, using a second order structural equation modelling for the ambidexterity variable.

This study also demonstrates the importance of the Big Five personality model for ambidexterity, especially when interpreting the results obtained through the Theory of Dynamic Capabilities and of micro-foundations associated with the owner-managers of these firms [93], as well as the Behavioural Theory of the Firm [7]. These theories sug-

gest relationships between firm performance and their owner-managers' personal traits. Therefore, it is argued that the owner-manager's personality traits affect how firms act and position themselves strategically in markets, and other studies over the years have demonstrated that relationship [94,95]. Therefore, the hypotheses formulated in this study sought to show that, also in an SME-specific context, owner-managers' personality traits can affect organisational ambidexterity.

This study also contributed to a number of questions still awaiting answers in the current literature. By placing this study at an individual level of analysis and seeking to contribute to a better understanding of personality traits and their influence on organisational ambidexterity in SMEs, it was possible to obtain a wider spectrum concerning the behaviour, tendencies, traits, and cognitions associated with owner-managers in an SME context. In fact, the facets associated with each factor influence organisational ambidexterity through the elements that act at both the exploration and exploitation level.

Despite the tendency to accept the notion that personality traits predict a great amount of organisational behaviour, a significant part of the literature considers the fact that not all the measures of personality traits have predictive power, especially when associated with self-reporting data-collecting instruments and the validity of measures associated with different organisational criteria [87].

For management, this study contributes in two ways: firstly, to situate the study of ambidexterity in SMEs as a primary antecedent, suggesting that certain personality traits, such as extraversion, can be considered adaptively for each different business environment in SMEs, as indeed suggested in some literature [96]. Secondly, this study contributes to the literature on entrepreneurship, since the results obtained allow for the proposal of different approaches to preparing new managers for Portuguese SMEs, focusing more on behaviour.

## 6. Limitations and Suggestions for Future Research

Future work can study the relationship of other intrinsic factors and their moderating roles, such as time leading the company, qualifications, and experience with owner-manager personality traits in organisational ambidexterity in SMEs. Other studies could be carried out applying this model to other sectors of activity. Qualitative studies could also be made over time in order to complement traditional data-collecting methods, such as questionnaires, and the measures of perception associated with these. For a better understanding of how the owner-manager's personality is connected to company performance, a multidisciplinary approach is necessary that would involve both the individual perspective and the structural and organisational perspective simultaneously. This idea is suggested by Hambrick and Mason [6], since, for these authors, the complexity of the phenomenon requires different scientific prisms of analysis.

This study presents several limitations. The first is related to the sector of activity chosen and its business context. Another limitation is related to the fact that this study considers the effect of external and internal characteristics in the relationship between personality traits and organisational ambidexterity.

**Author Contributions:** Conceptualization, J.A.; Methodology, J.A.; Investigation, J.A.; Writing—review & editing, J.A.; Supervision, L.M. and M.F. All authors have read and agreed to the published version of the manuscript.

**Funding:** This research was funded by Fundação para a Ciência e Tecnologia (UIDB/04007/2020).

**Data Availability Statement:** Data are contained within the article.

**Acknowledgments:** The authors would like to extend their sincere appreciation to the National Funds of the FCT—Portuguese Foundation for Science and Technology.

**Conflicts of Interest:** The authors declare no conflict of interest.

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
