# Peer review of "The Effect of Owner-Managers’ Personality Traits on Organisational Ambidexterity in the Context of Small and Medium-Sized Enterprises"

_sustainability, doi:10.3390/su16020507_

Round 1

Reviewer 1 Report

Comments and Suggestions for Authors

This manuscript explores the influence of the personality traits of owner-managers in small and medium-sized enterprises on organisational ambidexterity. This manuscript also studies the relationships between the Big Five personality traits and organisational ambidexterity. It is a topic of interest to the researchers in the related areas. which is within the scope of the journal, but there are some areas that need further modification. The paper has the potential to be accepted for publication. Before that, the authors are advised to consider the following comments and suggestions.

Comments:

1. Regarding the literature review on personality, the manuscript only outlines the concept of big-five model of personality. It is recommended to increase literature on the relationship between personality traits and company performance.

2. The description of the range of BFI-K in the manuscript is not clear, and it is recommended to further explain it.

3.Part of the data in Table 2 is obscured, it is recommended to change the format of Table 2.

4. It is recommended to change the color of the dashed line in Figure 1 so that it corresponds to the instructions.

5. The image in the text are unclear, such as Figure 1. It is recommended to further improve the quality of this image.

6. It is recommended to provide a more detailed explanation of the validation of the measurement model, including robustness testing.

7. The manuscript is long and lacks conciseness. It is recommended to make appropriate deletions from some lengthy expressions. For example, the description of the characteristics of the organisational ambidexterity can be appropriately reduced, as it has already been detailed in the introduction and literature review.

Author Response

We were pleased to know that, in general, reviewers find some merit in our research and considered that the issue covered in our paper could be suitable for publication and thus we thank the opportunity to review our paper for a potential publication in the journal. We recognize that the several observations and suggestions expressed by reviewers contributed to rise the paper’s quality level, and thus we appreciated all these contributions. In accordance, several changes have been performed, to improve the overall paper, in line with the different suggestions proposed.

  1. Regarding the literature review on personality, the manuscript only outlines the concept of big-five model of personality. It is recommended to increase literature on the relationship between personality traits and company performance. Literature relating personality traits to company performance has been added. L140 to L146
  2. The description of the range of BFI-K in the manuscript is not clear, and it is recommended to further explain it. We have tried to improve this aspect: L370 to L388
  3. Part of the data in Table 2 is obscured, it is recommended to change the format of Table 2. This aspect has been improved through minor adjustments to the table.
  4. It is recommended to change the color of the dashed line in Figure 1 so that it corresponds to the instructions. The color has been changed to black.
  5. The image in the text are unclear, such as Figure 1. It is recommended to further improve the quality of this image. The image has been improved, as has the information it contains.
  6. It is recommended to provide a more detailed explanation of the validation of the measurement model, including robustness testing. An explanation has been added. L478 to L484
  7. The manuscript is long and lacks conciseness. It is recommended to make appropriate deletions from some lengthy expressions. For example, the description of the characteristics of the organisational ambidexterity can be appropriately reduced, as it has already been detailed in the introduction and literature review. Several improvements have been made to the text: EXAMPLES: L41 to L47; L84 to L86; L110 to L113; L134 to L138; L148 to L158; L195 to 198; L223 to 226; L246 to 249 ; L326

Reviewer 2 Report

Comments and Suggestions for Authors

Review of the manuscript:

The Effect of Owner-Managers’ Personality Traits on Organisational Ambidexterity in SME’ Context

The subject of the manuscript generally corresponds to the purpose of the journal 'Sustainability'. Conclusions from data analysis after thorough revision of the manuscript may be helpful in the field of sciences related to the management of organizations and enterprises.

Unfortunately, I do not recommend the article in its current form for publication.

Justification of my opinion

1) The article is for the most part a collection (compilation) of the results of other authors’ work. I am not convinced by the methodology adopted.

2) No information on the type of manuscript.

3) Incorrect way of writing the names of Authors and their affiliations. Firstly, the names should be listed with the numerals next to them. Then, according to the designations (numbers), the affiliation of the Authors should be recorded.

4) Incorrect way of citing sources. Instead of names and year, the consecutive number of the cited literature should be given in square brackets.

For example: (Birkinshaw, Julian & Gupta, 2013) --> [1].

5) The References list should be arranged in the order in which they appear in the text, not in alphabetical order.

6) Careless preparation of the References list! The Authors have not considered the standards used in the MDPI publication. I provide an example below, which please consider as a model and please include as valuable literature in the discussion.

Kardas, J.S. Job Crafting and Work–Life Balance in A Mature Organization. Sustainability 2023, 15, 16089. https://doi.org/10.3390/su152216089.

7) Repeated References: No. 10 and No. 11; No. 34 and No. 35; No. 37 and No. 38; No. 52 and No. 53; No. 55 and No. 56; No. 66 and No. 67; No. 68 and No. 69

8) L610: Do not use such distinctions in the text:

Author Response

We were pleased to know that, in general, reviewers find some merit in our research and considered that the issue covered in our paper could be suitable for publication and thus we thank the opportunity to review our paper for a potential publication in the journal. We recognize that the several observations and suggestions expressed by reviewers contributed to rise the paper’s quality level, and thus we appreciated all these contributions. In accordance, several changes have been performed, to improve the overall paper, in line with the different suggestions proposed.

1) The article is for the most part a collection (compilation) of the results of other authors’ work. I am not convinced by the methodology adopted. 

2) No information on the type of manuscript. corrected. L14

3) Incorrect way of writing the names of Authors and their affiliations. Firstly, the names should be listed with the numerals next to them. Then, according to the designations (numbers), the affiliation of the Authors should be recorded. CORRECTED.

4) Incorrect way of citing sources. Instead of names and year, the consecutive number of the cited literature should be given in square brackets. CORRECTED

5) The References list should be arranged in the order in which they appear in the text, not in alphabetical order. CORRECTED

6) Careless preparation of the References list! The Authors have not considered the standards used in the MDPI publication. I provide an example below, which please consider as a model and please include as valuable literature in the discussion. CORRECTED.

Kardas, J.S. Job Crafting and Work–Life Balance in A Mature Organization. Sustainability 202315, 16089. https://doi.org/10.3390/su152216089.

7) Repeated References: No. 10 and No. 11; No. 34 and No. 35; No. 37 and No. 38; No. 52 and No. 53; No. 55 and No. 56; No. 66 and No. 67; No. 68 and No. 69

8) L610: Do not use such distinctions in the text: CORRECTED. L561 TO 578

Reviewer 3 Report

Comments and Suggestions for Authors

The authors of the paper titled "The effect of owner-managers’ personality traits on organisational ambidexterity in SME’ context" discussed a theoretical problem of the influence of the personality traits of owner-managers in small and medium-sized enterprises on organizational ambidexterity.

The paper contains a comprehensive literature review, but the titles are not cited correctly. I also suggest splitting the Literature Review and Research Hypotheses chapter into two. Reviewing the literature and presenting hypotheses are different things. It is also necessary to improve the text formatting, e.g. in line 121 there is unnecessary underlining and bolding of the text.

Most of the work is a kind of duplication and confirmation of research described in the literature. The paper is mainly theoretical considerations regarding the functioning of companies.

The response rate of 19% is relatively low; in my opinion, the sample should be increased. The analysis is very general. The analysis should be extended with additional research.

Why was the analysis limited to the sectors of information technology, telecommunications, audio-visual, and IT consulting?

The readability and formatting of Table 2 should be improved.

There are many typos. Moderate editing of the English language is required.

In my opinion, many corrections and major revisions are required to be able to reconsider and accept the paper. Please also consider changing the journal for the above paper because I am not sure that the topics discussed in the paper are suitable for the areas covered in Sustainability.

Comments on the Quality of English Language

There are many typos. Moderate editing of the English language is required.

Author Response

We were pleased to know that, in general, reviewers find some merit in our research and considered that the issue covered in our paper could be suitable for publication and thus we thank the opportunity to review our paper for a potential publication in the journal. We recognize that the several observations and suggestions expressed by reviewers contributed to rise the paper’s quality level, and thus we appreciated all these contributions. In accordance, several changes have been performed, to improve the overall paper, in line with the different suggestions proposed.

The paper contains a comprehensive literature review, but the titles are not cited correctly. I also suggest splitting the Literature Review and Research Hypotheses chapter into two. Reviewing the literature and presenting hypotheses are different things. It is also necessary to improve the text formatting, e.g. in line 121 there is unnecessary underlining and bolding of the text. CORRETED

Why was the analysis limited to the sectors of information technology, telecommunications, audio-visual, and IT consulting? The justification for choosing this type of company for this study lies in the fact that the market of the sampled SMEs is characterized by a high degree of uncertainty. In such a context, to be competitive, SMEs depend greatly on their capacity to adjust and adapt to new developments, circumstances, and demands. This adaptation and adjustment are typical mechanisms of exploration and exploitation, where organisational activities in-corporate the social and technical infrastructure of SMEs, contributing to organisational ambidexterity. L306 TO L312

The readability and formatting of Table 2 should be improved. CORRECTED.

There are many typos. Moderate editing of the English language is required. SOME CORRECTIONS WERE MADE.

In my opinion, many corrections and major revisions are required to be able to reconsider and accept the paper. Please also consider changing the journal for the above paper because I am not sure that the topics discussed in the paper are suitable for the areas covered in Sustainability.

Reviewer 4 Report

Comments and Suggestions for Authors

Dear Authors,

I had the opportunity to read and review the manuscript entitled „The effect of owner-managers’ personality traits on organisational ambidexterity in SME’ context”.

The aim of this study is to investigate organisational ambidexterity from the perspective of the big-five model of personality.

In my opinion, the research topic is absolutely in the scope of the journal's focus, the findings of this study help to better understand the factors affecting the innovation potential, resilience, and agility of SMEs. My review below suggests some improvements.

Title: informative and consistent with the study's content.

Abstract: the abstract is clear and reasonable.

Keywords: have to be reconsidered, a couple of them are already present in the title.

1. Introduction: the introduction provides a sufficient overview of the purpose of the study.

2. Literature Review and Research Hypotheses: this section provides a good overview of the current research on the topic, as well as presents the research dimensions and hypotheses. The hypotheses are supported by literature sources.

3. Research Methodology: the sampling procedure is clearly explained. A good explanation of how the scales were developed is also provided.

4. Results and Discussion: a validation of the structural and measurement models is provided. Research findings are well supported by the model results and discussed in comparison with the results of the previous studies.

5. Conclusions and Contributions: the conclusions are supported by the study findings.  The study provides practical implications for positioning ambidexterity as a primary antecedent of SMEs and for preparing managers for Portuguese SMEs with diverse personality traits

6. Limitations and Suggestions for Future Research: The limitations and suggestions for future research provided by the authors are acceptable to me.

It is my recommendation that this article be accepted for publication in its present form. There were no objectionable solutions or evaluations on my part. Study results established by the authors can be valuable both theoretically and practically.

Author Response

We were pleased to know that, in general, reviewers find some merit in our research and considered that the issue covered in our paper could be suitable for publication and thus we thank the opportunity to review our paper for a potential publication in the journal. We recognize that the several observations and suggestions expressed by reviewers contributed to rise the paper’s quality level, and thus we appreciated all these contributions. In accordance, several changes have been performed, to improve the overall paper, in line with the different suggestions proposed.

Reviewer 5 Report

Comments and Suggestions for Authors

A general discussion has been provided in the literature section on owner-manager personality and ambidexterity. It would be more convincing if the author(s) specifically mentioned how each trait affects explorative and exploitative innovations. Also, provide logical arguments for each hypothesis. For example, how emotional stability predicts exploitative and explorative innovations.

Revisit hypothesis H2, are you sure that owner-manager openness to experience is not related to organizational ambidexterity? As per the mentioned literature, it could be positively related to……visit open innovation framework. Also, revisit H4 for the same comment.  

The author(s) may visit affective event theory for personality traits.

All SMEs cannot reach the same level of ambidexterity, since exploitative and explorative innovations vary from SME to SME due to SME culture, structure their level of dynamic capabilities, and their openness to innovation. How accurately the findings of the study can be generalized, as the data in the present study has been collected from 1202 companies (line, 350, p.7)?

Provide details on, how randomization was done.

I didn’t find explanation on how modification was done on survey instruments (line 373). Provide details on item(s) modification with justification.

Justify your response rate of 19%.

How the construct of organizational ambidexterity was operationalized? As per information provided on page 9, line 441 and 569, it has been considered as a high-order construct. However, no information on the specification of measurement model under measurement theory has been provided. The author tested the structured model based on AMOS, this software lacks the property to measure the reflective formative second-order construct. What I can assume is that organizational ambidexterity has been conceptualized as a reflective-reflective second-order construct which is against the logic of measurement theory. see for details: Construct Measurement and Validation Procedures in MIS and Behavioral Research: Integrating New and Existing Techniques on JSTOR

Provide the structural model generated by AMOS.

The cited literature in the paper is outdated. I didn’t find literature from 2019-2023.

Provide valid justification on how openness to experience and agreeableness are not related to organizational ambidexterity. 

Comments on the Quality of English Language

Moderate edit requires 

Author Response

We were pleased to know that, in general, reviewers find some merit in our research and considered that the issue covered in our paper could be suitable for publication and thus we thank the opportunity to review our paper for a potential publication in the journal. We recognize that the several observations and suggestions expressed by reviewers contributed to rise the paper’s quality level, and thus we appreciated all these contributions. In accordance, several changes have been performed, to improve the overall paper, in line with the different suggestions proposed.

A general discussion has been provided in the literature section on owner-manager personality and ambidexterity. It would be more convincing if the author(s) specifically mentioned how each trait affects explorative and exploitative innovations. Also, provide logical arguments for each hypothesis. For example, how emotional stability predicts exploitative and explorative innovations. The arguments have been improved. L177 to L188

Revisit hypothesis H2, are you sure that owner-manager openness to experience is not related to organizational ambidexterity? As per the mentioned literature, it could be positively related to……visit open innovation framework. Also, revisit H4 for the same comment.  The literature presented in the study states that openness to experience is a personality trait characterized by the search for novelty, new experiences, and change, which are aspects related to exploration and not exploitation. Our reasoning is based on the idea that if this trait only supports exploration and not exploitation, then it does not support organizational ambidexterity, which was evidenced by the results obtained in this study. The arguments have been improved. L259 to L261

All SMEs cannot reach the same level of ambidexterity, since exploitative and explorative innovations vary from SME to SME due to SME culture, structure their level of dynamic capabilities, and their openness to innovation. How accurately the findings of the study can be generalized, as the data in the present study has been collected from 1202 companies (line, 350, p.7)? This study is innovative in that it seeks to establish a relationship between personality traits and organizational ambidexterity and shows that some of these personality traits influence organizational ambidexterity. Like all studies of this type, this one also has its limitations and the issue of sampling is one of them. However, we are convinced that this study is an important contribution to the literature.

Provide details on, how randomization was done. Improved: L302 to L315

I didn’t find explanation on how modification was done on survey instruments (line 373). Provide details on item(s) modification with justification. CORRECTED: L483 TO L490

How the construct of organizational ambidexterity was operationalized? IMPROVED/CORRECTED: L363 TO L375 AND L395 TO L402.

Provide the structural model generated by AMOS. IMPROVED: L505 TO L510

Justify your response rate of 19%. We consider that a response rate of 19% is quite acceptable and this is mentioned in the literature presented in the study.

The cited literature in the paper is outdated. I didn’t find literature from 2019-2023. Literature has been added corresponding to this period. 

Provide valid justification on how openness to experience and agreeableness are not related to organizational ambidexterity. The literature presented in the study states that openness to experience is a personality trait characterized by the search for novelty, new experiences, and change, which are aspects related to exploration and not exploitation. Our reasoning is based on the idea that if this trait only supports exploration and not exploitation, then it does not support organizational ambidexterity, which was evidenced by the results obtained in this study.

Reviewer 6 Report

Comments and Suggestions for Authors

1. The study aims to analyse the influence of the personality traits of owner-managers in 14 small and medium-sized enterprises (SME) on organisational ambidexterity.
2. The research is relevant.
3.  It provides an analysis of the personality traits.
4. The paper is written in a nice and correct manner.
5. The conclusions are consistent with the results provided in the paper.
6. The references are appropriate.

7. The paper has been written in a nice and detailed manner. Each subsection of it was analyzed in great detail.

All the best,

Author Response

(The authors gave the same response as above.)

Round 2

Reviewer 2 Report

Comments and Suggestions for Authors

The Authors followed the advice of the first review, carried out significant corrections.

On the other hand, they still haven't declared the manuscript type - Line 1.

Author Response

We appreciate the reviewer's comments. The type of manuscript is an empirical study. This information was placed in the abstract.

Reviewer 3 Report

Comments and Suggestions for Authors

I still have some concerns about the response rate of 19% mentioned in your research. In my opinion, this percentage is too low to draw any concrete conclusions or carry out further research on such a small sample size.

There is little current literature, most of it from before 2020 (3 items from 2022). I suggest updating the literature review on the examined issue. The literature review should be more detailed and the hypotheses justified.

Throughout the article, the term "model" is frequently used, but a detailed description of its construction and formulation is not provided. This makes it difficult to determine whether it has been used correctly.

Some of the comments have been taken into account, but I still believe that the paper has serious flaws, and additional experiments are needed.

Author Response

We appreciate the reviewers' suggestions and understand that these suggestions seek to raise the level of the paper. In this sense, our effort was to try to respond in the best possible way to the suggestions for improvement proposed by the reviewers.

I still have some concerns about the response rate of 19% mentioned in your research. In my opinion, this percentage is too low to draw any concrete conclusions or carry out further research on such a small sample size.

R-We are aware that the sample is small. However, literature was consulted that allowed us to make some important decisions regarding the determination of the sample size (n). We followed the recommendation of Westland (2010). However, the rule proposed by the author may vary depending on the specific research context, the type of SEM model, and the complexity of the model. Our research model is simple and not complex. Naturally, the number, metrics, and strength of the correlation between manifest variables, on the one hand, and the number of latent variables and structural relationships considered in the model, as well as the number of parameters to be estimated, on the other, are also determining factors for the required sample size. The "rule" proposed by Westland (2010) was applied and we considered the sample size to be adequate for this study.

Westland, J. C. (2010). Lower bounds on sample size in structural equation modeling. Electronic commerce research and applications9(6), 476-487.

There is little current literature, most of it from before 2020 (3 items from 2022). I suggest updating the literature review on the examined issue. The literature review should be more detailed and the hypotheses justified.

R-Additional relevant and current literature has been introduced into the paper. The hypotheses were improved with some additional literature and greater detail.

Throughout the article, the term "model" is frequently used, but a detailed description of its construction and formulation is not provided. This makes it difficult to determine whether it has been used correctly.

R-Some "model" terms were removed from the text. The construction of the model (validation of the measurement model and the structural model) is described in section 4. We do not understand this observation.

Some of the comments have been taken into account, but I still believe that the paper has serious flaws, and additional experiments are needed.

Reviewer 5 Report

Comments and Suggestions for Authors

As mentioned, the authors tested the structured model based on AMOS, this software lacks the property to measure the reflective formative second-order construct. What I can assume is that organizational ambidexterity has been conceptualized as a reflective-reflective second-order construct which is against the logic of measurement theory. Measurement model misspecification leads to type 1 and type II errors. consequently, these errors forbid researchers from adequate theory testing and meaningful theory development. 

My comment with reference to H2 is that if openness to experience is positively affecting explorative innovation how is possible that it will not lead to cosmetic changes? For example, if I am open to experience and can do explorative innovation (radical) then it is too easy for me to do incremental innovation (incremental). Therefore, I still disagree with H2.  

 The structural model provided is not the one generated by AMOS, not yet included.

 As I mentioned earlier, all SMEs cannot reach the same level of ambidexterity, since exploitative and explorative innovations vary from SME to SME due to SME culture, structure their level of dynamic capabilities, and their openness to innovation. How accurately the findings of the study can be generalized, as the data in the present study has been collected from 1202 companies? Justify the same level of ambidexterity in these 1202 companies.

The randomization process has not yet been addressed.

Explain how research instruments were modified before data collection, The information provided is about dropped items not about modification. Also, provide measurement instruments in the paper.

Authors opinions on updated literature are not included. I would suggest critically analyzing the literature. 

Author Response

We appreciate the reviewers' suggestions and understand that these suggestions seek to raise the level of the paper. In this sense, our effort was to try to respond in the best possible way to the suggestions for improvement proposed by the reviewers.

As mentioned, the authors tested the structured model based on AMOS, this software lacks the property to measure the reflective formative second-order construct. What I can assume is that organizational ambidexterity has been conceptualized as a reflective-reflective second-order construct which is against the logic of measurement theory. Measurement model misspecification leads to type 1 and type II errors. consequently, these errors forbid researchers from adequate theory testing and meaningful theory development.

Regarding this reviewer's observation, we can inform you that we collaborated with colleagues and involved specialized reviewers to review and validate the methodology and statistical analysis.

We evaluated the practical consequences of type II and type I errors in relation to the study context considering the theory that frames this study.

We assume that increasing the sample size is difficult at this time. We know this is one of the most significant factors in determining test power. Increasing the sample size generally increases the test's ability to detect statistically significant differences. This is a limitation of this study. As far as we know, there are no studies that have the same approach. In this sense, this study is exploratory and needs to be tested in other contexts.

My comment with reference to H2 is that if openness to experience is positively affecting explorative innovation how is possible that it will not lead to cosmetic changes? For example, if I am open to experience and can do explorative innovation (radical) then it is too easy for me to do incremental innovation (incremental). Therefore, I still disagree with H2.  

Regarding H2, It is possible that the subjects of this study could be characterized by not having this personality trait clearly defined in their personality. This aspect is understandable and can also help us understand the results obtained. We are dealing with a personality trait. For us, this H2 hypothesis makes sense. Other aspects can also help to understand this result and are mentioned in this study, namely that openness to experience may not act as a facilitator of organizational ambidexterity, mainly due to contextual factors, such as business strategy, company culture, difficulty in adopting innovation practices or lack of human and technical or financial resources. For us, it is not at all unreasonable to propose H2, considering the literature suggests on this topic. Check please section 2.2. and section 4.6. Discussion.

Hypotheses were rewritten.

The structural model provided is not the one generated by AMOS, not yet included.

Section 4 has a description of the model. We do not understand this point.

As I mentioned earlier, all SMEs cannot reach the same level of ambidexterity, since exploitative and explorative innovations vary from SME to SME due to SME culture, structure their level of dynamic capabilities, and their openness to innovation. How accurately the findings of the study can be generalized, as the data in the present study has been collected from 1202 companies? Justify the same level of ambidexterity in these 1202 companies.

Section 3.3. describes the approach used regarding this aspect of the level of organizational ambidexterity achieved by the companies in the study. Table 1 has been redesigned to facilitate understanding of this point.

The randomization process has not yet been addressed.

We do not understand this point.

Explain how research instruments were modified before data collection, The information provided is about dropped items not about modification. Also, provide measurement instruments in the paper.

If referring to the questionnaire, this point is answered with the introduction of table 2.

Authors opinions on updated literature are not included. I would suggest critically analyzing the literature.

We believe that this aspect has been improved with the introduction of more recent literature.

Notes for editor: Improvements were made throughout the text. Some improvements are marked in yellow.

Round 3

Reviewer 3 Report

Comments and Suggestions for Authors

Most of the comments have been taken into account in the new version of the manuscript at a level that allows consent for publication.

Author Response

We are grateful for the reviewer's suggestions.

Reviewer 5 Report

Comments and Suggestions for Authors

The current study is lacking in novelty. I noticed earlier that sections seem to be unoriginal, having appeared in already published work. The overlap of this submission with previously published articles is also too high for your article to be published in Sustainability.  Find the links below for details, and also check for H2.

https://doi.org/10.3390/su14010306

https://doi.org/10.1007/BF03396909

The cited paper (78) for personality measurement is not written in English, how this scale has been used? Similarly, I didn’t find any questionnaire form for ambidexterity in the cited literature (70 and 79).

How 1,202 Portuguese companies were selected randomly? The statement used to measure key variables in the study are questionable. For example; your action and I see myself?

Organizational ambidexterity in this study is misspecified. 

Comments on the Quality of English Language

Moderate English language required. 

Author Response

We are grateful for the reviewer's suggestions and we understand that these suggestions seek to raise the standard of the work. In this sense, our effort has been to try to respond as best we can to the suggestions for improvement proposed by the reviewer.

The current study is lacking in novelty. I noticed earlier that sections seem to be unoriginal, having appeared in already published work. The overlap of this submission with previously published articles is also too high for your article to be published in Sustainability.  Find the links below for details, and also check for H2.

  • We have made some changes to the structure of the study. We think that with these changes we have met the reviewer's request by reformulating some sections of the study.  All the suggested changes have been highlighted in the text in blue.

The cited paper (78) for personality measurement is not written in English, how this scale has been used? 

  • A better explanation of the scale used in this section has been introduced (Section Personality).  

Similarly, I didn’t find any questionnaire form for ambidexterity in the cited literature (70 and 79). 

  • The scale in the cited literature was requested directly from the authors.

How 1,202 Portuguese companies were selected randomly? 

  • The answer to this question can be found in the text, sampling section. 

The statement used to measure key variables in the study are questionable. For example; your action and I see myself? 

  • Sorry, we didn't understand that remark.

Organizational ambidexterity in this study is misspecified. 

  • We improved the definition of the concept of organizational ambidexterity with a better explanation of it and went a little further by looking for inconsistencies related to the translation of the concept throughout the text, which we didn't find.